# Exploring the drivers of price variation in orthopaedic radical bone tumor resection: A nationwide database study

Devika A. Shenoy[1], William C. Cruz[2], Shamik Bhat[3,4]*, Katelyn Parsons[1], Aaron D. Therien[1], Kevin A. Wu[5], Christian A. Pean[1], William C. Eward[1]

**1** Department of Orthopaedic Surgery, Duke University School of Medicine, Durham, North Carolina, United States of America, **2** School of Osteopathic Medicine, Campbell University, Durham, North Carolina, United States of America, **3** School of Public Policy, Princeton University, Princeton, New Jersey, United States of America, **4** School of Medicine, Yale University, New Haven, Connecticut, United States of America, **5** Department of Orthopedic Surgery, Icahn School of Medicine at Mount Sinai, New York City, New York, United States of America

* shamik.bhat@yale.edu

## Abstract

### Background

Radical resection of bone tumors is a clinically effective but costly procedure. Despite the implementation of federal price transparency mandates, little is known about the nationwide variation in negotiated prices for these specialized oncologic surgeries. This study aimed to quantify the variation in negotiated rates for radical resection of the humerus and femur/knee and identify associated hospital, payor, and state-policy drivers.

### Methods

This cross-sectional study analyzed hospital-negotiated payor rates from the Turquoise Health database for current procedural terminology (CPT) codes 24150 (humerus resection) and 27365 (femur/knee resection). Multivariate linear regression was used to determine the associations between hospital size and type, payor class, and state-level policies (Medicaid expansion, Certificate of Need [CoN] laws, All-Payer Claims Database [APCD] mandates, and Nurse Practitioner [NP] scope of practice) on negotiated payor rates.

### Results

A total of 285,857 negotiated rates were analyzed. Significant price variation was observed across all factors. Large hospitals (>1000 beds) and Critical Access Hospitals (for femur/knee resection only) had significantly higher rates. CoN laws were associated with higher prices for both procedures (+$348.25 and +$667.98, respectively), as were APCD mandates for femur/knee resections (+$1231.24). Medicare

**Data availability statement:** Payor information obtained from Turquoise Health: https://turquoise.health. Medicaid and regulatory information extracted from Kaiser Family Foundation and National Conference of State Legislatures: https://www.kff.org/affordable-care-act/state-indicator/state-activity-around-expanding-medicaid-under-the-affordable-care-act/?currentTimeframe=0&sortModel=%7B%22colId%22:%22Location%22,%22sort%22:%22asc%22%7D, https://www.ncsl.org/health/certificate-of-need-state-laws NP independent practice regulatory data from American Association of Nurse Practitioners Data: https://www.aanp.org/advocacy/state/state-practice-environment All-Payer Claims Database data from University of New Hampshire and National Association of Health Data Organizations: https://www.apcdcouncil.org/state-efforts/apcd-legislation-state Additional data availability information: 1. Negotiated rate (primary outcome), payor type, total bed range of hospital, hospital type, hospital location: Obtained from the Turquoise Health "Clear Rates" database under a licensing agreement. Further information can be found at https://turquoise.health/products/clear_rates_data. Researchers interested in working with this dataset can contact Turquoise Health via https://turquoise.health/contact?page=contact-us. 2. Medicaid expansion status: obtained from a publicly-available source via the Kaiser Family on December 31st, 2024: https://www.kff.org/affordable-care-act/state-indicator/state-activity-around-expanding-medicaid-under-the-affordable-care-act/?currentTimeframe=0&sortModel=%7B%22colId%22:%22Location%22,%22sort%22:%22asc%22%7D 3. Certificate of Need status: Data was obtained from a publicly-available source via the National Conference of State Legislatures on December 31st, 2024: https://www.ncsl.org/health/certificate-of-need-state-laws 4. NP scope-of-practice regulatory laws: Data was obtained from a publicly-available source via the American Association of Nurse Practitioners on July 1st, 2025: https://www.aanp.org/advocacy/state/state-practice-environment 5. All-Payer Claims Database (APCD) state participation: Data was obtained from a publicly-available source via the University of New Hampshire and National Association of Health Data Organizations on

Advantage plans paid inconsistently compared to commercial plans, paying more for humerus but substantially less for femur/knee resections.

## Discussion

Negotiated prices for radical bone tumor resection are highly variable and influenced by a complex interplay of market dynamics, challenging the assumption that price transparency alone can standardize healthcare costs for specialized care.

## Introduction

The management of primary and metastatic bone tumors has evolved greatly over the past few decades, with new options for advanced surgical reconstructions and targeted immunotherapy providing patients with multifaceted treatment options. [1–4] While primary bone cancers account for only 0.2% of primary malignancies in the United States (U.S.), approximately 5.1% of patients with other malignancies report developing bone metastases. [5,6] The national cost burden for patients with metastatic bone disease estimated at $12.6 billion dollars in 2007. [7] Despite advancements in medical therapies [8], surgical management remains at the cornerstone of ensuring local tumor control, and preserving access to high-quality surgical care is essential for patients. [9] For example, radical resection of bone tumors involves removing sections from long bones, often in the setting of primary bone tumors such as osteosarcoma, or for metastatic bone disease. [10–12] While clinically effective in providing local tumor control and preserving limb function [10–12], there is limited knowledge regarding the costs of these procedures in the past decade.

Due to their impacts on both direct patient costs and costs to the healthcare system, the financial dynamics of major surgical procedures are increasingly under scrutiny. [13–18] The recent implementation of the Centers for Medicare & Medicaid Services (CMS) Price Transparency Rule provides an opportunity to address these concerns. [19] By mandating that hospitals publicly disclose their privately negotiated rates with insurers, this policy has offered the ability to investigate the true, market-driven prices for specific procedures on a national scale, making it possible to explore the financial landscape of specialized oncologic surgeries. Recent studies leveraging price transparency data have uncovered significant, often unexplained, variation in the negotiated prices for common procedures in both general orthopaedics and surgical oncology. [14,15,17,18,20] For example, one cross-sectional study of 15,013 Medicare beneficiaries found large, regional variations in negotiated payor rates for common oncologic operations, as well as mild associations between high rates and adverse clinical outcomes. [18] Additionally, another study found that healthcare policies, such as Medicaid Expansion and Certificate of Need (CoN) status, were associated in large variations in payor rates for total hip arthroplasty. [14]

However, it is unknown if similar price variability exists for high-stakes, less common procedures like the radical resection of bone tumors. Therefore, the present study leverages a national price transparency database to analyze two analogous

July 1st, 2025: https://www.apcdcouncil.org/state-efforts/apcd-legislation-state 6. U.S. Census Bureau Region: Data was obtained from a publicly-available source via the U.S. Census Bureau at https://www2.census.gov/geo/pdfs/maps-data/maps/reference/us_regdiv.pdf.

**Funding:** The author(s) received no specific funding for this work.

**Competing interests:** The authors have declared that no competing interests exist.

**Abbreviations:** CPT = Current Procedural Terminology, CoN = Certificate of Need, APCD = All-Payer Claims Database, NP = Nurse Practitioner, CMS = Centers for Medicare & Medicaid Services, TQH = Turquoise Health, VA = Veteran Affairs, CAH = Critical Access Hospitals, SD = Standard Deviation, IQR = Interquartile Range.

but distinct procedures, radical resection of a tumor in the humerus and the femur or knee, the most common locations for such procedures. The objectives of this study were to quantify the nationwide variation in their negotiated prices along common hospital, payor, and state-policy level factors. While the specific policies analyzed are unique to the U.S., the findings on how market dynamics influence the valuation of complex surgical care have global implications for health systems navigating the challenges of cost containment and price transparency.

## Methods

### Overview

This is a cross-sectional study evaluating negotiated prices between hospitals and payors from the Turquoise Health (TQH) "Clear Rates" Database, which aggregates U.S. hospital-reported negotiated payor rates for common procedures by current procedural terminology (CPT) code. [21]

### Inclusion criteria

CPT codes evaluated by this study included 24150 (i.e., radical resection of tumor, shaft of humerus or distal humerus) and 27365 (i.e., radical resection of tumor, femur or knee). Data were analyzed in one overall cohort containing both CPT codes, and then individually within a "radical resection of humerus" and a "radical resection of femur/knee" cohort.

For this analysis, we only included rates from the TQH database that had complete payor and regional information. Following studies with similar methodologies [15,16,20,22], we then excluded outliers within the top and bottom 10% of all negotiated rates.

### Data source

Data was extracted from TQH on December 31st, 2024, under a licensing agreement, and included procedure rate, the type of payor, total bed range of the hospital (i.e., a proxy for hospital size), hospital type, and hospital location. Data on Medicaid expansion status and CoN regulations were extracted from the Kaiser Family Foundation and the National Conference of State Legislatures using 2024 data, and was extracted on December 31st, 2024. [23,24] CoN laws are intended to limit healthcare cost growth by restricting market entry and expansion; their association with negotiated prices in orthopaedics is a key area of continued research. [14–16] Scope of practice regulations for advanced practice providers were included as an additional proxy for market competition, which can influence overall procedural rates. For example, broader practice authority may expand access to conservative or postoperative care options, potentially impacting surgical case volumes or procedure demands. [25] Data on Nurse Practitioner (NP) independent practice regulations were extracted from the 2024 American Association of Nurse Practitioners data on July 1st, 2025. [26] Lastly, state-mandated All-Payer Claims Database (APCD), which mandates the reporting of all public and private health claims, are a direct price transparency mechanism that may have an impact

on negotiation. [27] Data on state participation in APCD initiatives were extracted from the University of New Hampshire and the National Association of Health Data Organizations database on July 1st, 2025. [27]

### Hospital and payor-level variables

Procedure rates are standardized in 2024 U.S. Dollars. Categories for the type of payor included commercial, managed Medicaid, Medicare Advantage, dual Medicare-Medicaid, exchange, Veteran Affairs (VA), and worker's compensation. Total bed range was used as a proxy for hospital size and refers to the approximate number of inpatient hospital beds. Total bed range was defined as a categorical variable with 6 options, as reported by TQH: 1–100, 100–300, 300–500, 500–1000, 1000–1500, 1500 + . Hospital type was reported by TQH. Critical Access Hospitals (CAHs) are hospitals designated by CMS to provide care within a geographic area without other hospitals in a 35-mile radius. [28] All other hospitals were designated as acute care hospitals.

### State policy variables

Medicaid expansion status, CoN regulations, and state-mandated APCD participation were all defined as binary variables (yes/no). NP scope-of-practice regulations were categorized into "full practice," "restricted practice," and "no practice."

### Regional variables

Nine regions were defined according to the U.S. Census Bureau using the state of the listed hospital: New England, Mid-Atlantic, East North Central, West North Central, South Atlantic, East South Central, West South Central, Mountain, and Pacific. [29] A full breakdown of states in each listed region is provided in the Appendix in (S1 Table).

### Statistical analysis

First, a descriptive analysis was conducted to summarize cohort characteristics using frequencies and percentages for categorical variables and means with standard deviation (SD) for continuous variables. Second, a multivariate linear regression was performed to determine the influence of hospital, payor, regional, and state policy factors on negotiated rates. All regressions control for every included variable in the study. Before model finalization, we verified the assumptions of linear regression, including the linearity of independent variables and the homoscedasticity and normality of residuals. Multicollinearity was assessed using Variance Inflation Factors (VIF): all included variables demonstrated a VIF < 5, suggesting no significant collinearity between predictors. Regression coefficients represent the mean difference in U.S. dollars between the evaluated variable and the reference group. Model performance was evaluated using the adjusted $R^2$ to determine the proportion of variance in negotiated rates explained by the independent variables. Results are visualized using forest plots. A p-value ≤ 0.05 was considered statistically significant for all analyses, which were conducted in R Version 4.4.1 (The R Foundation for Statistical Computing, Vienna, Austria).

### Ethics approval

This study includes publicly available data and negotiated payor rates from the Turquoise Health Database. Given that negotiated payor rates are general listed prices publicly posted by a hospital, these are not associated with patients or hospital encounters. Thus, this study was approved as exempt by our Institutional Review Board (#Pro00112768).

## Results

### Cohort details

A total of 285,857 negotiated payor rates were included for analysis (Table 1). This comprised 177,728 rates for radical resection of the humerus (CPT 24150) and 108,129 rates for radical resection of the femur/knee (CPT 27365). For the

**Table 1. Overview of Included Negotiated Payor Rates for Radical Resection of Bone Tumors.**

| Variable | CPT 24150[a] | CPT 27365[b] |
|---|---|---|
| Number of Payor Rates | 177,728 | 108,129 |
| Mean Payor Rate (SD) | $5,645.95 ($2,626.63) | $5,575.91 ($3,613.73) |
| *Total Bed Range of Hospital (N (%))* | | |
| 1 - 100 | 51,183 (28.8%) | 32,753 (30.3%) |
| 100 - 300 | 64,454 (36.3%) | 36,928 (34.2%) |
| 300 - 500 | 35,477 (20.0%) | 23,151 (21.4%) |
| 500 - 1000 | 22,148 (12.5%) | 11,634 (10.8%) |
| 1000 - 1500 | 1,731 (1.0%) | 1,095 (1.0%) |
| 1500 + | 2,735 (1.5%) | 2,568 (2.4%) |
| *Hospital Type (N (%))* | | |
| Acute Care | Acute Care | Acute Care |
| Critical Access | Critical Access | Critical Access |
| *Payor Class (N (%))* | | |
| Commercial | 100,590 (56.6%) | 79,224 (73.3%) |
| Dual | 1,369 (0.8%) | 259 (0.2%) |
| Managed Medicaid | 17,264 (9.7%) | 13,428 (12.4%) |
| Medicare Advantage | 52,635 (29.6%) | 13,172 (12.2%) |
| Veterans Affairs | 3,126 (1.8%) | 800 (0.7%) |
| Workers' Compensation | 2,744 (1.5%) | 1,246 (1.2%) |
| *U.S. Census Bureau Division* | | |
| Middle Atlantic | 40,233 (22.6%) | 29,566 (27.3%) |
| New England | 11,617 (6.5%) | 5,225 (4.8%) |
| East North Central | 30,973 (17.4%) | 9,355 (8.6%) |
| East South Central | 15,367 (8.6%) | 8,060 (7.5%) |
| Mountain | 6,060 (3.4%) | 9,753 (9.0%) |
| Pacific | 17,818 (10.0%) | 9,221 (8.5%) |
| South Atlantic | 30,906 (17.4%) | 14,729 (13.6%) |
| West North Central | 11,812 (6.6%) | 12,654 (11.7%) |
| West South Central | 12,942 (7.3%) | 9,566 (8.8%) |

Abbreviations: SD = Standard Deviation; CPT = Current Procedural Terminology Code; US = United States.

[a]CPT 24150 = Radical resection of tumor, shaft of humerus or distal humerus.

[b]CPT 27365 = Radical resection of tumor, femur or knee.

humerus cohort, the mean payor rate was $5,645.95 (SD: $2,626.63). For the femur/knee cohort, the mean payor rate was $5,575.91 (SD: $3,613.73). The highest percentage of rates came from the Middle Atlantic geographic region (40,233 (22.6%) for 24150; 29,566 (27.3%) for 27365).

Most rates for both procedures came from hospitals with 100–300 beds (36.3% for humerus, 34.2% for femur/knee). Commercial payors represented the largest payor class for both cohorts, accounting for 56.6% of humerus resection rates and 73.3% of femur/knee resection rates. Rates from acute care hospitals were most common, representing 93.8% of the humerus cohort and 87.4% of the femur/knee cohort. A full overview of the payor rate characteristics is available in Table 1.

## Radical resection of humerus cohort

A multivariate linear regression for the humerus resection cohort (CPT 24150) revealed several significant hospital-level factors associated with payor rates (Fig 1). Compared to hospitals with 1–100 beds, rates mostly increased with bed count

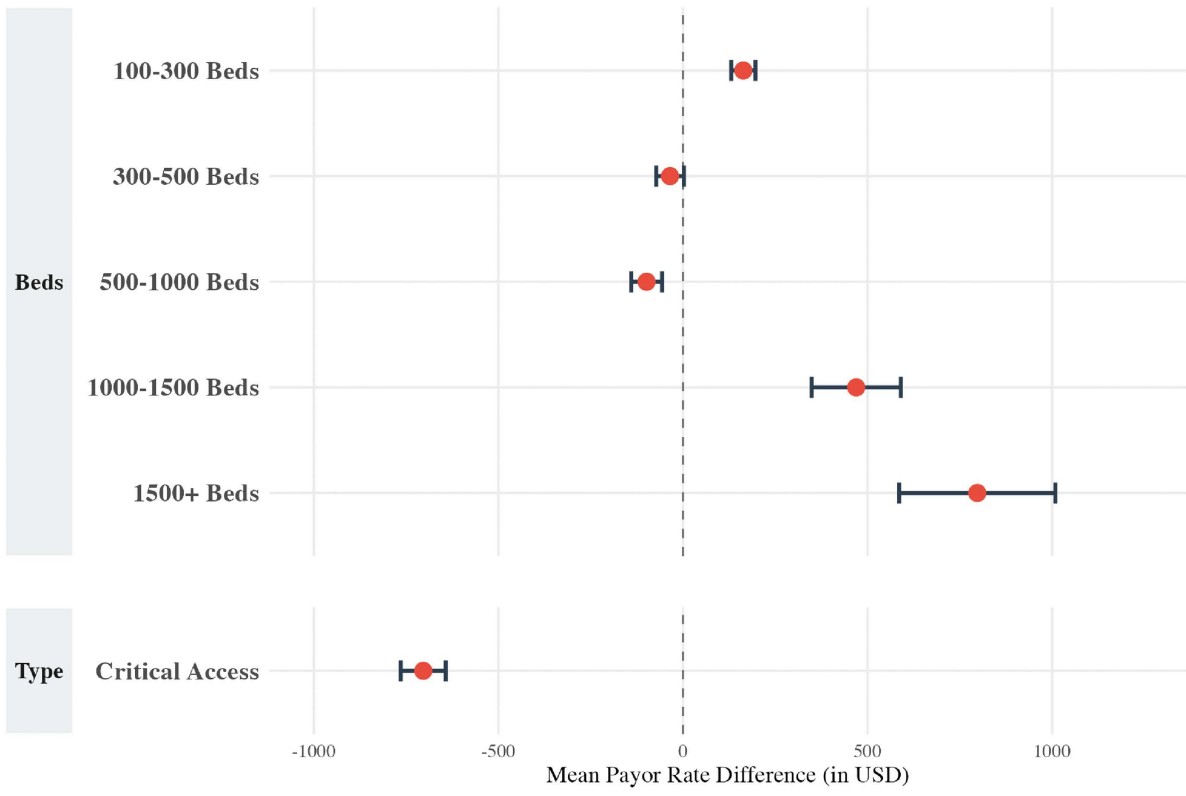

**Fig 1. Multivariable linear regression results for the humerus cohort, displayed as a forest plot showing the mean payor rate difference (in USD) for hospital size ("Beds") and hospital type ("Type").** The model controls for payor class and **U**.S. Census Bureau region. Reference groups include hospitals with 1–100 beds (for "Beds") and acute care facilities (for "Type").

and were the highest in facilities with 1500 + beds (+$797.46, p < 0.001). Conversely, rates were lower in hospitals with 500–1000 beds (-$98.48, p < 0.001). CAHs had rates that were $703.76 lower than acute care hospitals (p < 0.001).

Payor class was also associated with price variation (Fig 2). Relative to Commercial payors, rates were higher Workers' Compensation (+$1543.66, p < 0.001) and Medicare Advantage ($71.18, p < 0.001) plans. Rates for Managed Medicaid (-$2016.69, p < 0.001) and Veterans Affairs (-$295.97, p < 0.001) were lower compared to Commercial payors.

In the regional analysis, except for the East North Central region, all regions had higher rates in comparison to the Middle Atlantic region (Fig 3) varying between $172.04 higher (p < 0.001, East South Central) and $1,769.43 higher (p < 0.001, New England). All data for the humerus regression can be found in eAppendix in S2 Table.

A sub-analysis of health policies for the humerus resection cohort revealed that most evaluated policies were associated with higher mean payor rates compared to states without the policies (Fig 4). This included Medicaid expansion (+$274.64, p < 0.001), CoN laws (+$348.25, p < 0.001), and restricted NP practice (+$662.08, p < 0.001). On the other hand, the existence of an APCD mandate was not significantly associated with a change in rates (p = 0.97) and states with full NP practice had lower rates (-$226.29, p < 0.001). All data for the humerus cohort health policy sub-analysis can be found in eAppendix in S3 Table.

### Radical resection of femur/knee cohort

For the femur/knee resection cohort (CPT 27365), a multivariate linear regression identified several significant hospital-level variables impacting payor rates (Fig 5). The impact of hospital size was variable: most facility sizes were associated

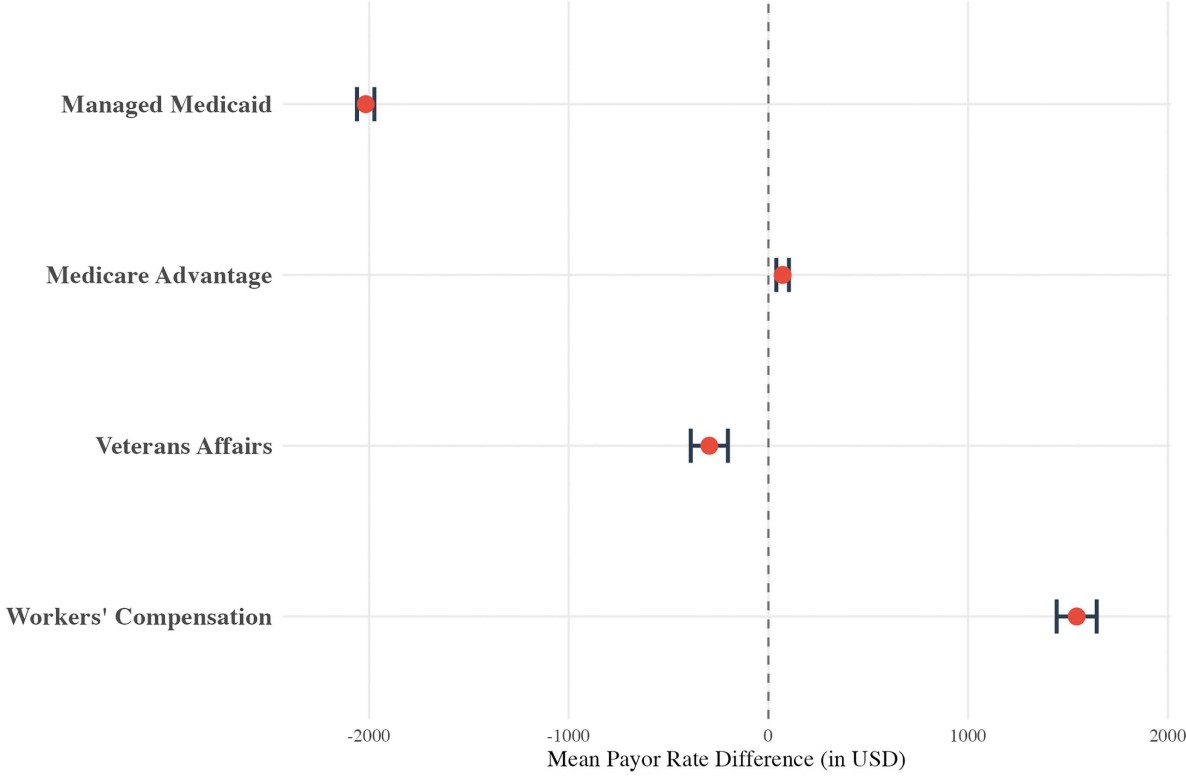

**Fig 2. Multivariable linear regression results for the humerus cohort, displayed as a forest plot showing the mean payor rate difference (in USD) for "Payor Class."** The model controls for hospital bed size, hospital type, and **U.**S. Census Bureau region. The reference group is Commercial.

with lower payor rates compared with hospitals with 1–100 beds, except hospitals with 1000–1500 beds (+$1753.12, p < 0.001) and 1500 + beds (+$3409.96, p < 0.001).

All payor classes had significantly lower rates when compared to the Commercial payor class (Fig 6), with the Medicare Advantage class having the lowest rates (-$2946.11, p < 0.001). The regional analysis also revealed significant variability (Fig 7): all regions had rates that were higher than the Middle Atlantic division, ranging from +$252.58 (p < 0.001, West South Central) to $3,791.18 (p < 0.001, South Atlantic). All data for the femur/knee regression can be found in eAppendix in S4 Table.

The health policy sub-analysis for the femur/knee cohort also found that most evaluated policies were associated with higher mean payor rates compared to states without such policies (Fig 8). This included Medicaid expansion (+$1399.41, p < 0.001), CoN laws (+$667.98, p < 0.001), restricted NP practice (+$1563.49, p < 0.001), and APCD mandates (+1231.24, p < 0.001). Conversely, rates were lower in states with full practice (-$250.79, p < 0.001). All data for the femur/knee cohort health policy sub-analysis can be found in eAppendix in S5 Table.

## Discussion

To our knowledge, this is the first study to leverage a national price transparency database to investigate the negotiated payor rates for radical resection of bone tumors. Our analysis of over 285,000 negotiated rates for radical resection procedures of the humerus and femur/knee reveals significant price variation driven by hospital, payor, regional, and state-level policy factors. These findings are similar to previous studies that identify large, often unexplained variations in negotiated payor rates. [14,15,17,18] For patients with primary or metastatic bone disease, who often require complex and costly

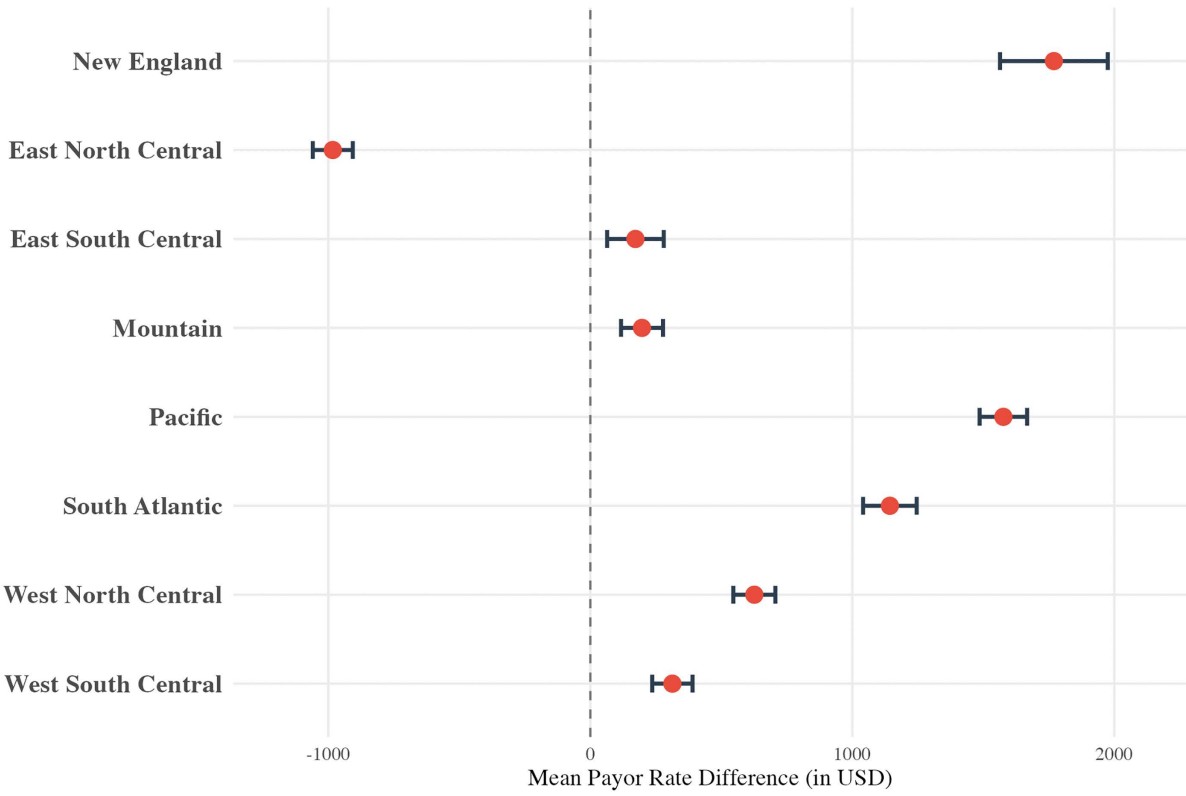

**Fig 3. Multivariable linear regression results for the humerus cohort, displayed as a forest plot showing the mean payor rate difference (in USD) for "U.S Census Bureau region."** The model controls for hospital bed size, hospital type, and payor class. The reference group is "Middle Atlantic.".

care, this variability raises important questions about health equity in a specialized field where patient choice may be limited by geography and clinical expertise.

A primary finding of this study is that hospital size is a significant driver of negotiated rates. For both procedures, the largest hospitals (1000 + beds) had substantially higher prices than smaller facilities. This is particularly pronounced for femur/knee resections, where rates at the very largest hospitals were approximately $3400 higher than at the smallest. This may reflect the market power of large hospitals, often with academic sarcoma centers that can handle complex cases and consequently may have greater leverage in negotiations with payors. [30,31] Given that most orthopaedic oncologists who perform these procedures routinely are located in academic medical centers, these findings may mean patients and health systems have no choice but to pay for the higher-cost procedures, effectively creating a price-inelastic market where transparency alone is unlikely to foster competition or reduce costs. Additionally, although prior studies have found that larger hospitals have higher prices than smaller hospitals [30,31], we observed a non-linear trend in the present study, with mid-sized hospitals (300–1000 beds) having inconsistently high or low rates compared to both the smallest and largest facilities. This may suggest a competitive disadvantage for these institutions, which may lack the negotiating power of large systems (e.g., those with academic sarcoma centers). However, when considering hospital type, CAHs had significantly lower rates for both procedures. The lower rate for the humerus procedure at CAHs may align with expectations of lower-case volume or complexity in rural settings. [32] Together, these findings suggest that market power, whether in the context of specialized expertise of large academic centers or the geographic monopoly of small rural CAHs, may be a driver of high prices for specialized orthopaedic oncology care.

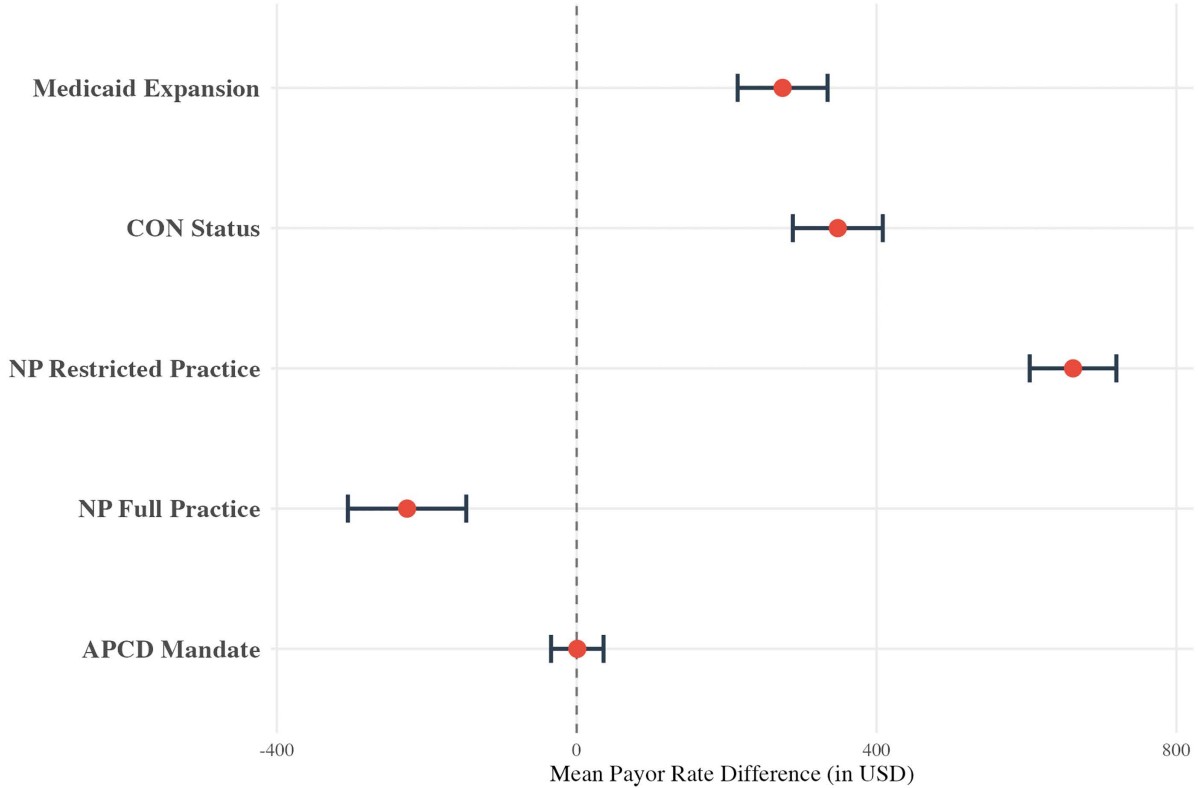

**Fig 4. Multivariable linear regression results for the humerus cohort, displayed as a forest plot showing the mean payor rate difference (in USD) for various health policies in states that do versus do not (reference) have the policy implemented.** Policies include Medicaid Expansion, Certificate of Need (CoN) status, nurse practitioner (NP) scope-of-practice laws, and All-Payer Claims Database (APCD) mandates. The model controls for hospital bed size, hospital type, payor class, and **U**.S. Census Bureau region.

Our analysis also uncovered regional differences in negotiated rates that persist even after controlling for hospital, payor, and state-policy factors. Using the Middle Atlantic (i.e., New York, New Jersey, Pennsylvania) as the reference, nearly all other U.S. regions had significantly higher prices for both procedures. The variation was particularly pronounced for femur/knee resections, where rates in the South Atlantic (e.g., North Carolina, South Carolina, Virginia) were nearly $3,800 higher, and rates in the West North Central region (e.g., North Dakota, South Dakota, Minnesota) were over $3,100 higher. Notably, the East North Central region was a significant outlier, with humerus resection rates nearly $1,000 lower than in the Middle Atlantic. Such large geographic variations may reflect underlying differences in regional market concentration, the negotiating leverage of dominant hospital systems in certain areas, or cost-of-living differences not fully captured by other variables.

The influence of payor type also revealed significant variability. Medicaid plans were associated with the lowest payor rates for both procedures. This finding is consistent with prior literature in orthopaedics, and may reflect limits on Medicaid reimbursement. [33,34] While lower rates are likely beneficial to patients with reduced cost-sharing, it is also important to ensure lower prices do not disincentivize providers from offering the service. For example, in neurosurgical populations, lower reimbursement rates for common procedures were associated with lower provider enrollment. [35] Future studies should evaluate access to orthopaedic oncology care in populations with Medicaid insurance to ensure that such patients are not facing additional barriers out of the scope of this study. Other insurance subtypes also showed unclear variability. For humerus resections, Medicare Advantage plans had higher rates than commercial plans (+$71), while for femur/

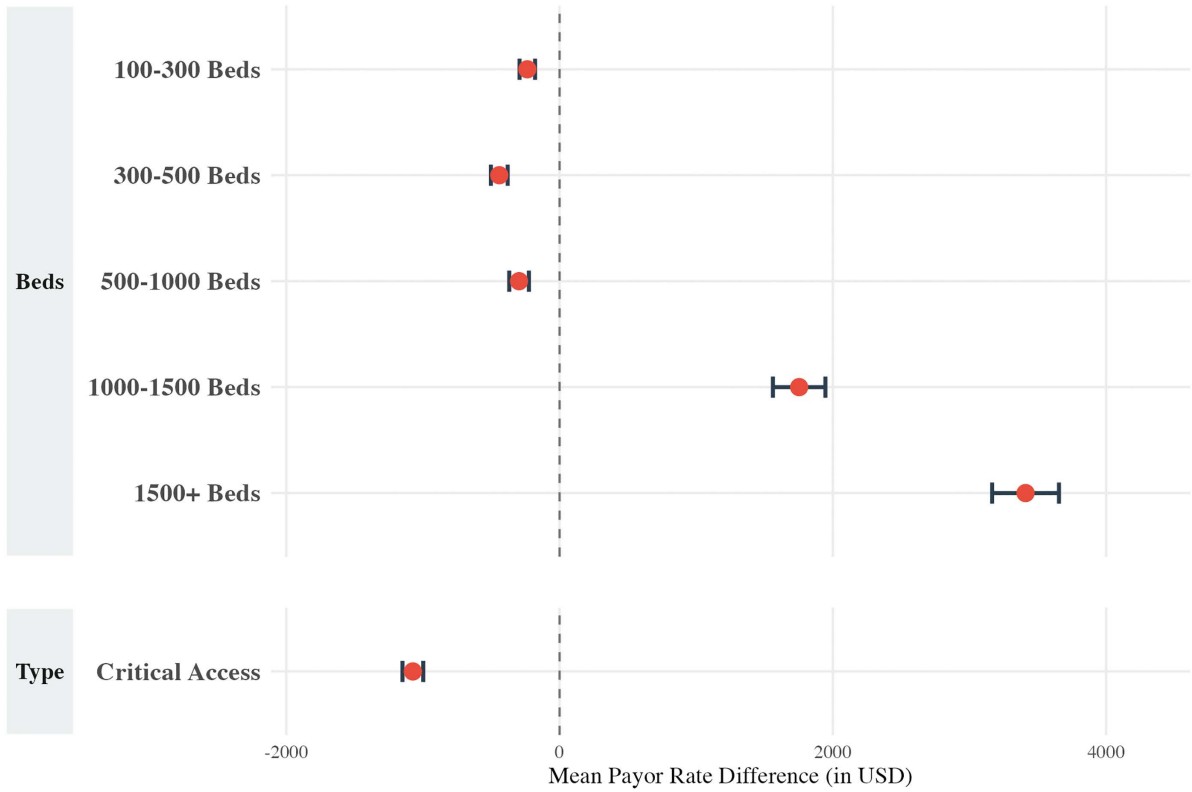

**Fig 5. Multivariable linear regression results for the femur/knee cohort, displayed as a forest plot showing the mean payor rate difference (in USD) for hospital size ("Beds") and hospital type ("Type").** The model controls for payor class and **U**.S. Census Bureau region. Reference groups include hospitals with 1–100 beds (for "Beds") and acute care facilities (for "Type").

knee resections, they paid significantly less (-$2,946). These differences suggest that the balance of power between a given hospital and a given Medicare Advantage plan can lead to vastly different prices for similar services. Taken together, these wide variations by insurer reveal a variable pricing system where the cost of essential cancer surgery is determined not only by clinical value, but also by the specific negotiating power and policies of a patient's insurance plan. However, whether this variability translates to differences in patient outcomes is a critical, yet unknown, question in orthopaedic oncology. In one gastrointestinal surgical cancer population, Sankaran et al found that despite high rate variation between similar providers, the median payor rate was not associated with mortality or major postoperative complications. [18] Future studies are needed to understand the relationship between payor-driven variation in procedure costs, access to care, and outcomes for orthopaedic oncology patients.

At the state policy level, our findings were often counterintuitive. The presence of CoN laws, which are intended to control costs by limiting market entry and ensuring a healthcare "need", were associated with significantly higher prices for both procedures. This finding conflicts with other studies in orthopaedic surgery, that found associations between CoN laws and lower prices. [13,15,16] In the context of orthopaedic oncology care, which already may face higher prices due to hospital-level factors, it is possible that CoN laws may inadvertently grant preexisting hospitals greater market power, allowing them to negotiate higher rates. Because the market for radical bone tumor resection is already limited to a small number of tertiary care centers with the requisite multidisciplinary expertise and infrastructure, CoN laws may effectively shield these centers from competition. This may further increase prices and limit access to care for patients who need radical resections.

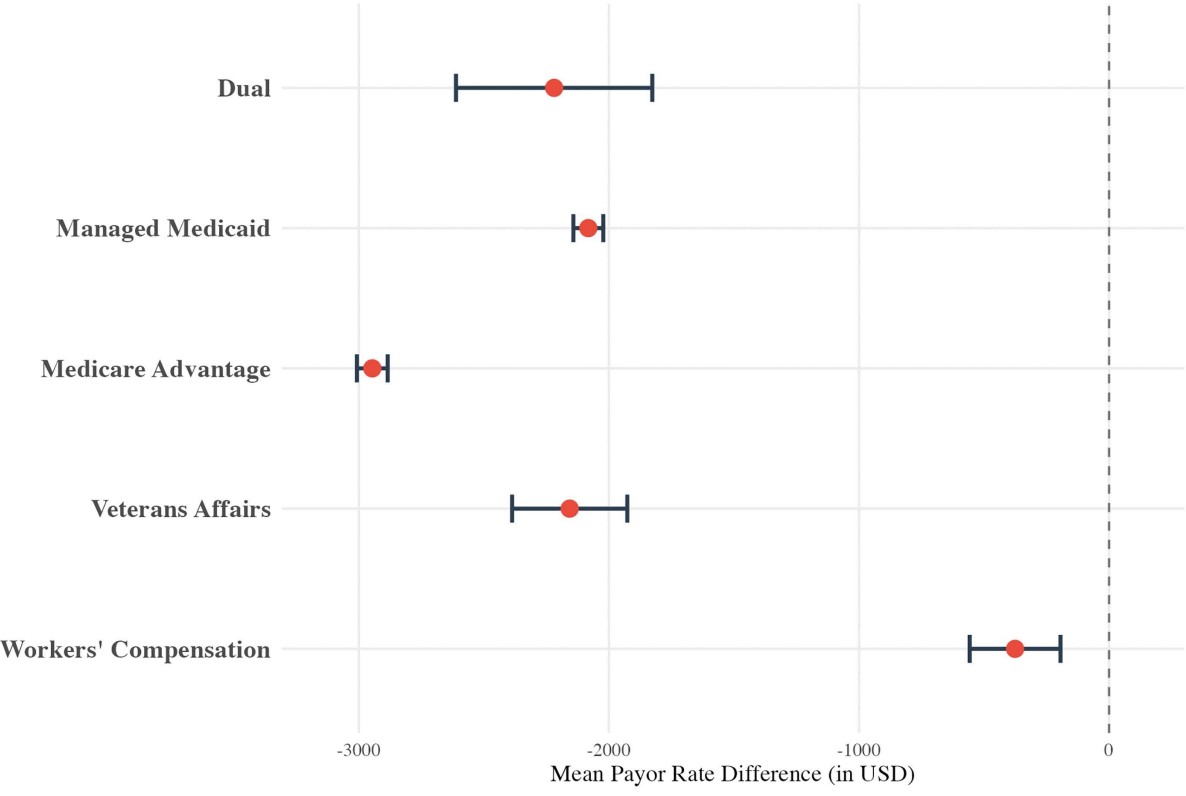

**Fig 6. Multivariable linear regression results for the femur/knee cohort, displayed as a forest plot showing the mean payor rate difference (in USD) for "Payor Class. "** The model controls for hospital bed size, hospital type, and **U**.S. Census Bureau region. The reference group is Commercial.

Medicaid expansion was linked to higher negotiated rates, consistent with prior research. [13,15,16] However, state-mandated APCDs, a direct price transparency mechanism, were not associated with humerus resection payor rates but were associated with $1,231 higher payor rates for femur/knee resections. In the case of radical resection of bone tumors, this contradicts the logic that transparency should foster competition and lower prices. This paradox may be specific to high-acuity, low-volume procedures where patient choice is driven by clinical expertise rather than cost. For example, a patient with a bone sarcoma may decide where to seek care based on the reputation of the surgical team and the geographic feasibility, rather than by comparing listed prices. [36] In this price-inelastic market, transparency may not empower patients to "shop" for lower prices, but instead may allow hospitals to adjust their own rates upward toward those of their high-cost peers knowing that patients will be restricted to their institutions, leading to an inflationary effect. Thus, in the radical bone tumor resection market, it is possible that transparency may allow lower-priced providers to increase prices to the market price, which may be inflated at baseline by the state- and hospital-level factors mentioned above.

The findings for NP scope of practice laws have implications for understanding the role of advanced practice providers in specialized surgical markets. For humerus resections, broader NP practice authority was linked to lower surgical rates, supporting the theory that an expanded workforce could increase competition or access to non-operative care. [25] Yet, for femur/knee resections, the opposite was true, with broader NP practice linked to significantly higher rates. This discrepancy may indicate that the role of advanced practice providers differs by procedure complexity and should be explored in future research. It is possible that these differences reflect differences in NP role based on tumor location. In the lower extremity, resections are often more likely to be essential due to lower extremity weight bearing requirements. For patients

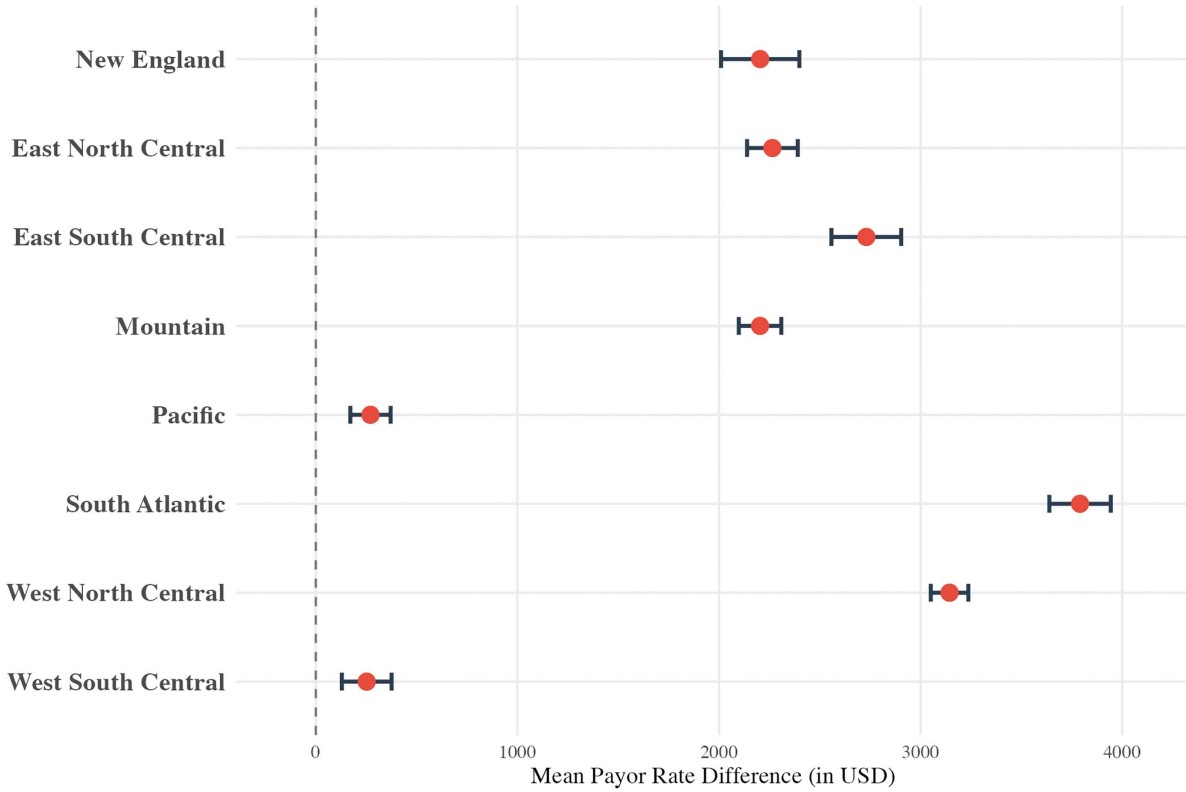

**Fig 7. Multivariable linear regression results for the femur/knee cohort, displayed as a forest plot showing the mean payor rate difference (in USD) for "U.S Census Bureau region. "** The model controls for hospital bed size, hospital type, and payor class. The reference group is "Middle Atlantic.".

with upper extremity tumors, is possible that access to NPs can result in alternatives to surgery, such as injections and counseling about weight bearing limitations. On the contrary, the same access to NPs for patients with lower extremity lesions could result in more rapid facilitation of a surgical intervention.

While this analysis is specific to the multi-payer healthcare system of the U.S., health systems worldwide, whether single-payer or market-based, are grappling with the challenge of controlling costs for specialized oncologic care, while ensuring access and quality. [37–39] For example, even in universal healthcare systems (e.g., Canada) where the government is permitted to negotiate the price of cancer treatments with manufacturers and direct patient-facing medical costs are lower, patients still experience high levels of financial distress related to their cancer care. [40,41] Our findings serve as a case study for variable market dynamics, demonstrating that well-intentioned policies like market entry regulation (CoN laws) and price transparency mandates can have counterintuitive, and even inflationary, effects on prices. For international policymakers considering market-based reforms or greater price disclosure, this study highlights that simply revealing prices or limiting provider supply does not guarantee cost containment. Further efforts are needed to determine the best way to limit costs. The complex interplay between provider access, regulatory frameworks, and negotiating power can subvert policy goals, which should be kept in mind when establishing fair and predictable pricing for complex medical procedures.

This study has several limitations. Most importantly, the prices included in this study account for the resection portion of the procedure, but they do not account for any reconstructive portion(s). In clinical practice, reconstruction is an integral and often more resource-intensive component of limb-sparing surgery than the resection itself. The reconstructive portion

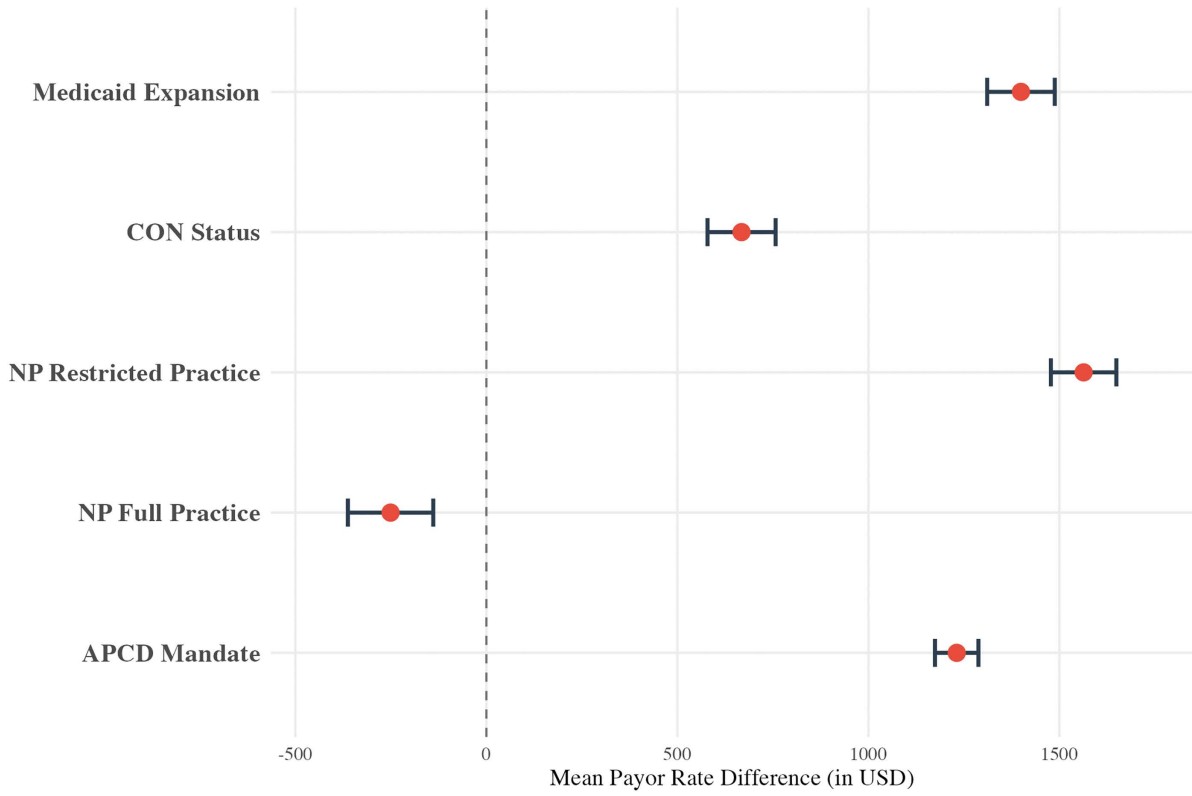

**Fig 8. Multivariable linear regression results for the femur/knee cohort, displayed as a forest plot showing the mean payor rate difference (in USD) for various health policies in states that do versus do not (reference) have the policy implemented.** Policies include Medicaid Expansion, Certificate of Need (CoN) status, nurse practitioner (NP) scope-of-practice laws, and All-Payer Claims Database (APCD) mandates. The model controls for hospital bed size, hospital type, payor class, and U.S. Census Bureau region.

may also have its own significant price variability associated with it due to market forces that drive implant prices. Consequently, the price variability observed in our study may not be representative of the complete episode of care associated with the surgical treatment of bone tumors. Future studies should incorporate all elements of bone tumor resection surgery to develop a complete cost profile for patients undergoing these procedures, including grafts, implants, and custom endoprostheses. Second, data were extracted from TQH, relying on the accuracy of each institution's disclosed negotiated rates. Hospitals or insurers may report only partial or grouped costs, potentially underestimating or overestimating total procedure expenses. Similarly, an additional limitation of this data source is that it includes rates from all hospitals, irrespective of whether they actually perform these complex procedures. Many smaller hospitals may list a negotiated rate for these CPTs but never perform the operation. This is particularly relevant for the interpretation of findings related to small hospitals and Critical Access Hospitals, whose negotiated rates may be theoretical rather than based on actual clinical volume. Third, this analysis only considers negotiated rates and does not account for variations in true reimbursement rates or direct out-of-pocket costs incurred by patients. Fourth, this study did not incorporate clinical outcomes (e.g., surgical complication rates) or patient-reported outcomes (e.g., financial toxicity or quality-of-life measures), which would be necessary to gauge the true value and cost-efficacy of each intervention in the long-term. Finally, our analysis was limited to two CPT codes for radical resection of long bones, and the findings may not be generalizable to other oncologic procedures or anatomical locations. While this approach was chosen to allow for a direct evaluation of two common procedures, future studies should investigate whether these pricing dynamics hold true for other surgical oncologic procedures.

## Conclusions

In conclusion, the price of radical bone tumor resection in the U.S. shows significant, often counterintuitive variability and may be driven in part by market dynamics. These findings underscore the urgent need for more granular data to understand these price drivers, which is essential for developing a more rational, predictable, and equitable pricing system for vital cancer care.

## Supporting information

**S1 Table. State Divisions According to United States Census Bureau.**
(DOCX)

**S2 Table. Multivariable Linear Regression for Payor Rates within the Radical Resection of Humerus Cohort.**
(DOCX)

**S3 Table. Multivariable Linear Regression for Payor Rates within the Radical Resection of Humerus Cohort, Health Policy Sub-Analysis.**
(DOCX)

**S4 Table. Multivariable Linear Regression Results for Payor Rates within the Radical Resection of Femur/Knee Cohort.**
(DOCX)

**S5 Table. Multivariable Linear Regression Results for Radical Resection of Femur/Knee Cohort, Health Policy Sub-Analysis.**
(DOCX)

## Author contributions

**Conceptualization:** Devika A. Shenoy, William C. Cruz, Christian A. Pean, William C. Eward.

**Data curation:** Devika A. Shenoy, William C. Cruz, Katelyn Parsons, Aaron D. Therien.

**Formal analysis:** Devika A. Shenoy, William C. Cruz, Katelyn Parsons, Aaron D. Therien.

**Investigation:** Kevin A. Wu.

**Methodology:** Devika A. Shenoy, William C. Cruz, Shamik Bhat, Kevin A. Wu, Christian A. Pean.

**Project administration:** Devika A. Shenoy, Christian A. Pean.

**Supervision:** Kevin A. Wu, Christian A. Pean, William C. Eward.

**Validation:** Shamik Bhat.

**Writing – original draft:** Devika A. Shenoy, William C. Cruz, Katelyn Parsons.

**Writing – review & editing:** Devika A. Shenoy, Shamik Bhat.

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
