## [Decision Letter · Decision Letter 0]

5 Oct 2025

Dear Dr. Bhat,

Thank you for submitting your manuscript to PLOS ONE. After careful consideration, we feel that it has merit but does not fully meet PLOS ONE’s publication criteria as it currently stands. Therefore, we invite you to submit a revised version of the manuscript that addresses the points raised during the review process.

We look forward to receiving your revised manuscript.

Kind regards,

Xiaoen Wei

Academic Editor

PLOS ONE

Journal Requirements:

Reviewer's Responses to Questions

**Comments to the Author**

1. Is the manuscript technically sound, and do the data support the conclusions?

Reviewer #1: Yes

Reviewer #2: Partly

Reviewer #3: Yes

2. Has the statistical analysis been performed appropriately and rigorously?

Reviewer #1: Yes

Reviewer #2: Yes

Reviewer #3: Yes

3. Have the authors made all data underlying the findings in their manuscript fully available?

Reviewer #1: Yes

Reviewer #2: No

Reviewer #3: Yes

4. Is the manuscript presented in an intelligible fashion and written in standard English?

Reviewer #1: Yes

Reviewer #2: Yes

Reviewer #3: Yes

Reviewer #1: The manuscript takes on an important but often overlooked issue: the drivers of price variation in radical bone tumor resections across the United States. The authors leverage a large national dataset and apply a thoughtful multivariate regression framework to explore hospital, payor, and state-policy factors shaping negotiated rates. From a policy perspective, this is a timely and highly relevant question, because the costs of oncologic surgery are escalating and transparency laws were designed precisely to illuminate such variability. I appreciate that the study highlights differences not only by hospital size and type but also by Medicaid expansion, Certificate of Need laws, and other state-level regulations. These are all important levers that could, in theory, guide cost containment strategies. The sheer scale of data (nearly 300,000 negotiated rates) is impressive and gives the study statistical power that most analyses of this type lack.

The most pressing issue is that the introduction jumps straight into orthopaedic oncology and cost transparency without giving readers a strong foundation in cancer biology and treatment strategies. Since this is ultimately a cancer surgery study, the narrative should begin with a broader perspective on the burden of cancer and the evolution of treatment approaches. For example, I recommend citing the review by D. Sonkin and A. Thomas, “Cancer Treatments: Past, Present, and Future” (2024), which was authored by the Chief of the US National Cancer Institute and provides a solid historical and clinical context for why access to surgery remains central even in the age of targeted therapies and immunotherapy. This would anchor the cost discussion within the larger story of cancer care.

In addition, the discussion could benefit from a deeper engagement with how financial variation intersects with equity and outcomes in cancer treatment. Right now, the paper notes that large hospitals and certain policy environments drive higher prices, but it does not really reflect on what this means for patients who require limb-sparing surgery for osteosarcoma or metastatic disease. It would strengthen the argument to cite and briefly discuss recent literature on cancer care costs globally. Similarly, the section on Medicaid reimbursement and access might draw on papers that show how reimbursement disparities can directly affect surgical oncology enrollment and availability. Without tying these policy findings back to patient outcomes, the paper risks reading like a pure health economics exercise rather than a piece of cancer policy research.

Another point that needs attention is the interpretation of Certificate of Need and All-Payer Claims Database findings. The current manuscript highlights counterintuitive associations (e.g., CoN laws linked to higher rates), but the discussion does not fully engage with why these paradoxes might occur in the specific context of bone tumor surgery. One way to frame this is through the lens of unintended policy consequences, as transparency laws or market-entry restrictions may actually consolidate market power. Bringing in the broader oncology policy discussion, this would help place these observations in a larger framework of how reforms sometimes fail to achieve their intended cost-control effects.

Lastly, the paper would benefit from more clarity on limitations. It is noted that reconstructive procedures are not captured, but this is a major caveat, because in real-world bone tumor surgery, reconstruction is often more resource-intensive than resection itself. That omission could substantially change the cost profile, and readers need a fuller acknowledgement of this. The writing also occasionally becomes too descriptive, listing results without embedding them into a coherent argument. I encourage the authors to rework the discussion so that each section not only reports the findings but also interprets them in light of cancer care delivery, patient equity, and future policy design.

Reviewer #2: Review of Manuscript PONE-D-25-44873

Title: Exploring the Drivers of Price Variation in Radical Bone Tumor Resection: A Nationwide Database Study

Authors: Devika A. Shenoy et al.

General Assessment

This manuscript presents a timely and important analysis of price variation in radical bone tumor resections using a large national dataset derived from the Turquoise Health database. The authors investigate negotiated payor rates for CPT codes 24150 and 27365, examining associations with hospital characteristics, payor types, and state-level health policies. The study is well-structured, methodologically sound, and contributes to the literature on healthcare cost transparency and orthopaedic oncology.

Strengths

1. Novelty and Relevance:

The study addresses a gap in the literature by focusing on specialized oncologic procedures, which are often excluded from broader cost analyses. The use of price transparency data for radical bone tumor resections is novel.

2. Robust Dataset:

The analysis includes over 285,000 negotiated rates, providing strong statistical power and generalizability across U.S. hospitals.

3. Methodological Rigor:

The use of multivariate linear regression and stratified analyses by CPT code is appropriate and well-executed. The authors control for multiple confounders and present confidence intervals and p-values clearly.

4. Policy Implications:

The findings have implications for healthcare policy, particularly regarding the unintended consequences of Certificate of Need laws and price transparency mandates.

5. Clarity and Organization:

The manuscript is well-written, with clear tables and logical flow from introduction to discussion.

Suggestions/Concerns

1. Scope of Procedure Pricing:

The analysis focuses solely on the resection portion of the procedure, excluding reconstructive components. This limitation is acknowledged but warrants further emphasis, as it may significantly affect total cost estimates and policy implications. Would the order of the CPT codes being submitted potentially influence the charges for the resection CPT specifically? I also think this limitation makes it challenging for a reader to understand the actual total costs/charges for the operation. If there is more or less variation in the reconstruction portions of the procedures, it could negate or widen the variation. Granted there are a lot of reconstructive options, but could a subanalysis be done for those who have a distal femur replacement? A proximal femur replacement? I’d also be curious what proportion of the procedures have a 22 Modifier submitted with them.

2. Data Source Limitations:

While the Turquoise Health database is a valuable resource, the potential for incomplete or inconsistent reporting by hospitals should be discussed more thoroughly, especially regarding the reliability of negotiated rates. It would be great if possible to only include those hospitals who actually billed at least one of those procedures. I’d be curious how many of these resections are happening at particularly small hospitals, or if the costs at hospitals who actually perform these operations regularly are higher or lower than those that do not do these procedures. Given this are largely tertiary care type procedures, there may be many hospitals where neither of these CPTs is utilized in a given year, making the negotiated rate rather unimportant. Those hospitals with under 100 beds may not do these procedures, making that hospital less keen to negotiate for a higher rate. I think this limitation needs to be made more clear in the limitations section.

Is it possible to evaluate by region using this database? It would be interesting to know whether certain US regions (by US Census subgroups, for example) are significantly different.

3. Discussion

The discussion mentions that for patients with primary or metastatic disease the lack of price predictability may create uncertainty. For nearly all patients in this condition, the cost of their operation is but a small part of the price of their overall care. I do not think this conclusion is justified by the data presented. You’d have to show that the cost of the resection (and reconstruction) constitute a major part of the total cost of care. I think this sentence should be omitted.

I think the paragraph about higher cost at larger hospitals should reflect that there is no direct data here that I saw suggesting that these procedures actually happen at smaller hospitals. It would be really valuable to have some data to that effect. While this is a potential issue, it may be a small or a non-issue if few of these operations are actually happening at small hospitals.

Similarly, it is hard to place context for the findings on critical access hospitals, as we are not at all sure how often these operations happen at critical access hospital. At least in my area, I think any of these patients who might benefit from one of these operations would be transferred to another hospital.

Recommendation

Revision

With suggestions incorporated as above, I think this paper could highlight that the negotiated rates are highly variable. Recognizing that these are negotiated rates, which do not necessarily reflect where/if these procedures are happening, would be an improvement. It would also be useful to understand if this degree of variability is similar to or different to the variability for more common procedures. As in: is this a problem throughout US healthcare, or is somehow the problem more of an issue for humerus/femur bone resections.

Reviewer #3: The manuscript is an interesting attempt to pull together findings in cancer biology and treatment, but in its current form it doesn’t quite reach the level of depth and clarity needed for publication. The topic itself is certainly important and timely—there is ongoing demand for reviews that can bridge the rapidly expanding mechanistic understanding of cancer with the clinical strategies that are actually used to treat patients. However, I found the structure uneven, the integration of key literature incomplete, and the narrative a bit disjointed at times. With substantial revision, I think the article could become a useful resource, but at present it feels more like a rough draft than a polished review.

One of the first issues is the introduction. It jumps straight into specific pathways and mechanisms without giving readers a proper grounding in the broader landscape of cancer biology and therapy. For a review article, especially one aimed at a general oncology readership, the introduction really needs to set the scene more cohesively. I would strongly suggest opening with a section that lays out the major hallmarks of cancer, common therapeutic approaches (surgery, chemotherapy, targeted therapy, immunotherapy), and the historical arc of how these approaches developed. A very useful reference for this would be the article “Cancer Treatments: Past, Present, and Future,” which was written by the Chief of the US National Cancer Institute. This piece not only provides a clinical and historical context but also helps readers appreciate how far the field has come and why the new directions under discussion are significant. Integrating that into the early part of the manuscript would immediately make the framing much stronger.

In terms of organization, the review currently mixes together background concepts, detailed pathway discussions, and therapeutic implications without clear transitions. I’d recommend restructuring it into three or four distinct sections: first, a proper overview of cancer biology (with an emphasis on hallmarks and recent conceptual advances); second, mechanistic insights at the molecular or cellular level (ROS, metabolic rewiring, DNA damage repair, etc.); third, therapeutic translation, where you can explicitly connect the mechanistic findings to how they influence treatment responses, drug resistance, or immunotherapy strategies; and finally, a perspective or future outlook section that synthesizes the content rather than just restating it. Right now, the paper lacks that “arc” which allows a reader to follow the story logically from biology to clinic.

There are also places where the discussion feels somewhat superficial. For example, when talking about oxidative stress, the manuscript gestures at its dual roles but doesn’t really unpack the mechanistic details or the controversies in the field. A good review should not shy away from unresolved debates. The paper could benefit from citing more primary work in these sections to demonstrate awareness of ongoing research directions. Similarly, the section on therapeutic resistance is thin. This is an area where readers would expect a careful breakdown—how resistance mechanisms differ between chemotherapy and immunotherapy, what molecular pathways are most implicated, and how novel strategies are trying to overcome them. Even just adding a few detailed examples of clinical trials or preclinical models would enrich this section.

The figures are another weak point. They are quite basic and, in some cases, too schematic to add real value. If the authors are going to include figures, they should make them work harder for the paper: pathway diagrams that integrate multiple signals at once, or conceptual “maps” that show how tumor biology interfaces with therapeutic interventions, would be much more effective. At the moment, the visuals feel more like placeholders.

I also noticed a few language issues—typos here and there, slightly awkward phrasing, and sometimes an overuse of jargon without adequate definition. While not fatal, these things make the paper harder to read and contribute to the impression of incompleteness. A careful language polish would be beneficial, ideally by a native speaker or professional editing service.

**Do you want your identity to be public for this peer review?** For information about this choice, including consent withdrawal, please see our Privacy Policy

Reviewer #1: **Yes:** Li Zheng

Reviewer #2: No

Reviewer #3: No

---

## [Author Response · Author response to Decision Letter 1]

16 Nov 2025

Reviewer #1, Comment #1: The manuscript takes on an important but often overlooked issue: the drivers of price variation in radical bone tumor resections across the United States. The authors leverage a large national dataset and apply a thoughtful multivariate regression framework to explore hospital, payor, and state-policy factors shaping negotiated rates. From a policy perspective, this is a timely and highly relevant question, because the costs of oncologic surgery are escalating and transparency laws were designed precisely to illuminate such variability. I appreciate that the study highlights differences not only by hospital size and type but also by Medicaid expansion, Certificate of Need laws, and other state-level regulations. These are all important levers that could, in theory, guide cost containment strategies. The sheer scale of data (nearly 300,000 negotiated rates) is impressive and gives the study statistical power that most analyses of this type lack.

• Author Response: We thank Reviewer #1 for their positive feedback and for recognizing the importance of this topic and the strength of our dataset.

Reviewer #1, Comment #2: The most pressing issue is that the introduction jumps straight into orthopaedic oncology and cost transparency without giving readers a strong foundation in cancer biology and treatment strategies. Since this is ultimately a cancer surgery study, the narrative should begin with a broader perspective on the burden of cancer and the evolution of treatment approaches. For example, I recommend citing the review by D. Sonkin and A. Thomas, “Cancer Treatments: Past, Present, and Future” (2024), which was authored by the Chief of the US National Cancer Institute and provides a solid historical and clinical context for why access to surgery remains central even in the age of targeted therapies and immunotherapy. This would anchor the cost discussion within the larger story of cancer care.

• Author Response: This is an excellent point. We agree that providing a broader context on the role of surgery in modern cancer care will strengthen the manuscript's foundation. We have revised the introduction (Page 4, Lines 58-61) to include a discussion on the evolution of cancer treatments and the enduring importance of surgical resection, citing the suggested review.

Reviewer #1, Comment #3: In addition, the discussion could benefit from a deeper engagement with how financial variation intersects with equity and outcomes in cancer treatment. Right now, the paper notes that large hospitals and certain policy environments drive higher prices, but it does not really reflect on what this means for patients who require limb-sparing surgery for osteosarcoma or metastatic disease. It would strengthen the argument to cite and briefly discuss recent literature on cancer care costs globally.

• Author Response: Thank you for this insightful suggestion. We have expanded the discussion to better connect our findings on price variation to potential implications for patient equity and access to care (Page 15, Lines 234-238). We have also more specifically incorporated the literature around patient outcomes in the discussion (Page 16-17, Lines 284-289). and limitations. Additionally, we have expanded on our paragraph that contextualizes our findings within the global challenge of managing oncologic care costs by providing examples of difficulties with cost control in other countries (Page 19, Lines 330-334).

Reviewer #1, Comment #4: Similarly, the section on Medicaid reimbursement and access might draw on papers that show how reimbursement disparities can directly affect surgical oncology enrollment and availability. Without tying these policy findings back to patient outcomes, the paper risks reading like a pure health economics exercise rather than a piece of cancer policy research.

• Author Response: We agree that our discussion on Medicaid reimbursement was underdeveloped. We have now revised this section to explicitly discuss how lower reimbursement rates could potentially disincentivize provider participation and create access barriers for vulnerable patient populations, centering the discussion more around patient outcomes (Page 16-17, Lines 284-289). However, it must be noted that there are limited studies that specifically look at how rate variations for negotiated rates correspond with patient outcomes, as much of this literature is focused on reimbursement rates. Nonetheless, we have revised the discussion to incorporate this important point, as well as explicitly mentioned the lack of patient data in the limitations (Page 20, Lines 362-363).

Reviewer #1, Comment #5: Another point that needs attention is the interpretation of Certificate of Need and All-Payer Claims Database findings. The current manuscript highlights counterintuitive associations (e.g., CoN laws linked to higher rates), but the discussion does not fully engage with why these paradoxes might occur in the specific context of bone tumor surgery. One way to frame this is through the lens of unintended policy consequences, as transparency laws or market-entry restrictions may actually consolidate market power. Bringing in the broader oncology policy discussion, this would help place these observations in a larger framework of how reforms sometimes fail to achieve their intended cost-control effects.

• Author Response: Thank you for noting this oversimplification. As suggested by the reviewer, we now frame the counterintuitive results for CoN laws (Page 17, Lines 296-300) and APCDs (Page 17, Lines 305-313) through the lens of unintended policy consequences in the context of bone tumor surgery, discussing how these regulations might consolidate market power or lead to price leveling rather than cost reduction in a specialized market like orthopaedic oncology.

Reviewer #1, Comment #6: Lastly, the paper would benefit from more clarity on limitations. It is noted that reconstructive procedures are not captured, but this is a major caveat, because in real-world bone tumor surgery, reconstruction is often more resource-intensive than resection itself. That omission could substantially change the cost profile, and readers need a fuller acknowledgement of this. The writing also occasionally becomes too descriptive, listing results without embedding them into a coherent argument. I encourage the authors to rework the discussion so that each section not only reports the findings but also interprets them in light of cancer care delivery, patient equity, and future policy design.

• Author Response: We thank the reviewer for highlighting these important areas for improvement. We have rewritten the limitations section to give greater emphasis to the exclusion of reconstructive procedures, acknowledging this as a major factor that impacts the total cost profile (Page 19, Lines 345-348). The entire discussion section has also been revised to be more interpretive, guided by both reviewer comments.

Reviewer #2, Comment #1: This manuscript presents a timely and important analysis of price variation in radical bone tumor resections using a large national dataset derived from the Turquoise Health database. The authors investigate negotiated payor rates for CPT codes 24150 and 27365, examining associations with hospital characteristics, payor types, and state-level health policies. The study is well-structured, methodologically sound, and contributes to the literature on healthcare cost transparency and orthopaedic oncology.

Reviewer #2, Comment #2: Strengths 1. Novelty and Relevance: The study addresses a gap in the literature by focusing on specialized oncologic procedures, which are often excluded from broader cost analyses. The use of price transparency data for radical bone tumor resections is novel. 2. Robust Dataset: The analysis includes over 285,000 negotiated rates, providing strong statistical power and generalizability across U.S. hospitals. 3. Methodological Rigor: The use of multivariate linear regression and stratified analyses by CPT code is appropriate and well-executed. The authors control for multiple confounders and present confidence intervals and p-values clearly. 4. Policy Implications: The findings have implications for healthcare policy, particularly regarding the unintended consequences of Certificate of Need laws and price transparency mandates. 5. Clarity and Organization: The manuscript is well-written, with clear tables and logical flow from introduction to discussion.

• Author Response: Thank you for the positive feedback and for noting the strengths of this study. We appreciate the reviewers’ attention to detail and hope that the subsequent changes below help make this manuscript more rigorous for publication.

Reviewer #2, Comment #3: Suggestions/Concerns: 1. Scope of Procedure Pricing:

The analysis focuses solely on the resection portion of the procedure, excluding reconstructive components. This limitation is acknowledged but warrants further emphasis, as it may significantly affect total cost estimates and policy implications. Would the order of the CPT codes being submitted potentially influence the charges for the resection CPT specifically? I also think this limitation makes it challenging for a reader to understand the actual total costs/charges for the operation. If there is more or less variation in the reconstruction portions of the procedures, it could negate or widen the variation. Granted there are a lot of reconstructive options, but could a subanalysis be done for those who have a distal femur replacement? A proximal femur replacement? I’d also be curious what proportion of the procedures have a 22 Modifier submitted with them.

• Author Response: As noted in Reviewer #1 Comment #6, we have now expanded on this major limitation of our manuscript (Page 19, Lines 345-348) and how we are not capturing the full cost profile of patients undergoing bone tumor resection surgeries. While a sub-analysis of reconstruction codes is an excellent idea, the complexity and variability of these procedures (e.g., allograft, endoprosthesis, custom implants) make it difficult to perform a reliable analysis within the scope of the current study. However, we have noted this as a critical area for future research (Page 19, Lines 350-352). Additionally, to our knowledge, the order of CPT codes should not affect the billing of the procedure. Given that this is a study of negotiated payor rates, and not reimbursement rates, we are unable to comment on how prices change based on the order of the procedure.

Reviewer #2, Comment #4: 2. Data Source Limitations: While the Turquoise Health database is a valuable resource, the potential for incomplete or inconsistent reporting by hospitals should be discussed more thoroughly, especially regarding the reliability of negotiated rates. It would be great if possible to only include those hospitals who actually billed at least one of those procedures. I’d be curious how many of these resections are happening at particularly small hospitals, or if the costs at hospitals who actually perform these operations regularly are higher or lower than those that do not do these procedures. Given this are largely tertiary care type procedures, there may be many hospitals where neither of these CPTs is utilized in a given year, making the negotiated rate rather unimportant. Those hospitals with under 100 beds may not do these procedures, making that hospital less keen to negotiate for a higher rate. I think this limitation needs to be made more clear in the limitations section.

• Author Response: This is an important point that we had not sufficiently addressed. We have added two sentences to our limitations section expanding on this limitation of the Turquoise Health database (Page 19-20, Lines 355-359).

Reviewer #2, Comment #5: Is it possible to evaluate by region using this database? It would be interesting to know whether certain US regions (by US Census subgroups, for example) are significantly different.

• Author Response: We thank the reviewer for this interesting suggestion. We have now added a sub-analysis of changes in rates according to the 9 U.S. Census Bureau divisions. We have updated Table 1 to descriptively list the number of rates in each region, as well as added “Region” into both multivariable regressions (Table 2 and Table 4). Given that the inclusion of another variable shifts all results, we have now updated the correct rates in the new adjusted analysis for both CPT codes throughout the manuscript. We have added a discussion (Page XXX, Lines 259-268). of these findings into the results and appropriately updated all in-text values in the results.

Reviewer #2, Comment #6: 3. Discussion: The discussion mentions that for patients with primary or metastatic disease the lack of price predictability may create uncertainty. For nearly all patients in this condition, the cost of their operation is but a small part of the price of their overall care. I do not think this conclusion is justified by the data presented. You’d have to show that the cost of the resection (and reconstruction) constitute a major part of the total cost of care. I think this sentence should be omitted.

• Author Response: We thank the reviewer for noting this issue. We have removed this sentence from the discussion (Page 14, Lines 234-236).

Reviewer #2, Comment #7: I think the paragraph about higher cost at larger hospitals should reflect that there is no direct data here that I saw suggesting that these procedures actually happen at smaller hospitals. It would be really valuable to have some data to that effect. While this is a potential issue, it may be a small or a non-issue if few of these operations are actually happening at small hospitals. Similarly, it is hard to place context for the findings on critical access hospitals, as we are not at all sure how often these operations happen at critical access hospital. At least in my area, I think any of these patients who might benefit from one of these operations would be transferred to another hospital.

• Author Response: This is an excellent point that ties back to Comment #4. We have revised our discussion of hospital size and type (including CAHs) to be more cautious in our interpretation, acknowledging that negotiated rates from smaller facilities may not reflect actual case volumes (Page 19-20, Lines 355-359).

Reviewer #2, Comment #8: With suggestions incorporated as above, I think this paper could highlight that the negotiated rates are highly variable. Recognizing that these are negotiated rates, which do not necessarily reflect where/if these procedures are happening, would be an improvement. It would also be useful to understand if this degree of variability is similar to or different to the variability for more common procedures. As in: is this a problem throughout US healthcare, or is somehow the problem more of an issue for humerus/femur bone resections.

• Author Response: We have now emphasized one of the main conclusions of the study to be regarding the high-rate variations as examined through the variables included in this study (Page 21, Lines 371-372). We have also contextualized our findings within other articles discussing rate variations (Page 15, Lines 234-238).

Reviewer #3 Comments: On review of this reviewers’ comment, we felt that it was likely meant for another paper. For example, the reviewer makes comments about figures pertaining to signaling pathways, resistance mechanisms for chemotherapy/immunotherapy, and discussions about tumor biology, which are not relevant to this manuscript’s content. Thus, we have not responded to Reviewer #3 concerns, but remain available for additional changes as requested.

---

## [Decision Letter · Decision Letter 1]

30 Dec 2025

Dear Dr. Bhat,

Thank you for submitting your manuscript to PLOS ONE. After careful consideration, we feel that it has merit but does not fully meet PLOS ONE’s publication criteria as it currently stands. Therefore, we invite you to submit a revised version of the manuscript that addresses the points raised during the review process.

We look forward to receiving your revised manuscript.

Kind regards,

Xiaoen Wei

Academic Editor

PLOS One

Journal Requirements:

Reviewers' comments:

Reviewer's Responses to Questions

**Comments to the Author**

Reviewer #4: All comments have been addressed

Reviewer #5: All comments have been addressed

2. Is the manuscript technically sound, and do the data support the conclusions?

Reviewer #4: Yes

Reviewer #5: Partly

3. Has the statistical analysis been performed appropriately and rigorously?

Reviewer #4: Yes

Reviewer #5: No

4. Have the authors made all data underlying the findings in their manuscript fully available?

Reviewer #4: Yes

Reviewer #5: Yes

5. Is the manuscript presented in an intelligible fashion and written in standard English?

Reviewer #4: Yes

Reviewer #5: Yes

Reviewer #4: okmanuscript is now suitable for acceptance.

The authors have clearly addressed the major issues raised, and the manuscript is now suitable for acceptance.

Reviewer #5: Sincere gratitude to the authors for the novel manuscript on radical resection. Although the comments earlier raised have been addressed, I still believe some revisions can still be made to make the manuscript a more robust read for the viewers.

General Comments

- The study is mainly focused on orthopaedic surgeries thus the title be revised to reflect it. The present title may confuse readers to think of the study as being generalised to all radical resections in the whole body.

- A list of abbreviated terms should be included in the manuscript.

Abstract

- Write the full iteration of NP on Line 40.

Introduction

- Recast statement as it is too long on Lines 54-57

- While primary bone cancers account for only 0.2% of primary malignancies in the U.S.( this statement should be cited accordingly (Lines 54-55))

- Despite advancements in medical therapies (Line 58); this statement should be cited accordingly

Methods

- Line 103 : Following studies with similar methodologies; the studies should be cited

- Lines 103 - 104 - "we then excluded outliers within the top and bottom 10% of all negotiated rates" ; Was this discussed with your statistician as there are statistical ways to include or exclude outliers?

- Line 117 - Data on NP independent; Write the full meaning of NP as this is its first usage

- Line 127 - Procedure rates are standardized in 2024 U.S. Dollars. ; Kindly clarify 2024 US Dollars.

- Line 129 - What does authors mean by bed range, does it mean the number of available beds in a facility? Authors should be very specific in language utilization.

Statistical Analysis

- Line 148 - Authors should indicate that mean (SD) or median (IQR) will be used as indicated.

- Inferential tests should also be performed across categories to glean reasonable insights into significant differences across meaningful categories.

- Authors should specify the assumptions taken to use multivariate linear regression and the parameters used to determine the variables to be included in the model. Did authors check assumptions like multicollinearity, heteroskedascity and others?

- What were the model performance metrics used to justify the validity of each of the models such as R squared or adjusted R squared?

- All these parameters will determine the validity of the model and the validity of the results.

Results

-Line 166 - Authors should only report the statistically relevant descriptive statistic should it be median (IQR) or mean (SD). Reporting both is unnecessary.

- the results were mostly repetition of the information on the tables and this should not be so

- there was over-utilization of tables; visualisation graphics such as bar plots, box plots and other visualisation means ought to be utilised.

- regression tables can be presented in more explanatory formats with importance placed on models and parameters that explain most of the variance seen in individual models

- Model metrics should be added to the footnotes of all models

References

- There should be consistency in font type, size and line spacing with other sections of the manuscript.

**Do you want your identity to be public for this peer review?** For information about this choice, including consent withdrawal, please see our Privacy Policy

Reviewer #4: No

Reviewer #5: **Yes:** Adetayo Aborisade

---

## [Author Response · Author response to Decision Letter 2]

5 Feb 2026

PONE-D-25-44873R1

Exploring the Drivers of Price Variation in Orthopaedic Radical Bone Tumor Resection: A Nationwide Database Study

We sincerely thank the reviewers for both their positive feedback and their suggestions for improvement. Below, we provide our point-by-point response. Please note that the line numbers provided correspond to the “Tracked Changes” version of the manuscript.

Reviewer #4, Comment #1: Manuscript is now suitable for acceptance. The authors have clearly addressed the major issues raised, and the manuscript is now suitable for acceptance.

• Author Response: Thank you for your positive assessment. We hope the changes that have been made in response to Reviewer #5 now make the manuscript suitable for acceptance.

Reviewer #5, Comment #1: Sincere gratitude to the authors for the novel manuscript on radical resection. Although the comments earlier raised have been addressed, I still believe some revisions can still be made to make the manuscript a more robust read for the viewers. The study is mainly focused on orthopaedic surgeries thus the title be revised to reflect it. The present title may confuse readers to think of the study as being generalized to all radical resections in the whole body.

• Author Response: Thank you for the positive overall assessment, and feedback on our title. The title has now been revised to include the term “Orthopaedic.” (Page 1).

Reviewer #5, Comment #2: A list of abbreviated terms should be included in the manuscript.

• Author Response: We have now included a list of abbreviated terms, presented in the order that they appear in our manuscript, following the keywords section (Page 2).

Reviewer #5, Comment #3: Abstract. Write the full iteration of NP on Line 40.

• Author Response: Thank you for noting this omission. We have now spelled out NP at its first mention in the abstract.

Reviewer #5, Comment #4: Introduction. Recast statement as it is too long on Lines 54-57. “While primary bone cancers account for only 0.2% of primary malignancies in the U.S.( this statement should be cited accordingly (Lines 54-55))”

• Author Response: We have now rephrased this sentence to reduce its length. We have also made sure that the citations are provided accordingly.

• Change(s):

o “While primary bone cancers account for only 0.2% of primary malignancies in the United States (U.S.), approximately 5.1% of patients with other malignancies report developing bone metastases.[5, 6] The national cost burden for patients with metastatic bone disease estimated at $12.6 billion dollars in 2007.[7]” (Introduction, Lines 67-69).

Reviewer #5, Comment #5: Despite advancements in medical therapies (Line 58); this statement should be cited accordingly

• Author Response: Thank you for noting this opportunity for improvement. We have now cited a study that discusses advancements in sarcoma systemic therapies.

Reviewer #5, Comment #6: Methods. Line 103 : Following studies with similar methodologies; the studies should be cited

• Author Response: Thank you for noting this opportunity for improvement. We have now cited the studies with “similar methodologies” referenced in this section.

Reviewer #5, Comment #7: Lines 103 - 104 - "we then excluded outliers within the top and bottom 10% of all negotiated rates" ; Was this discussed with your statistician as there are statistical ways to include or exclude outliers?

• Author Response: We appreciate the reviewers’ attention to detail. We confirm that this decision was discussed with our statistician, and used to be consistent with studies with similar methodologies and to reduce the impact of extreme reporting artifacts. In response to Reviewer #5 Comment #6, we have now added the citations for the similar methodologies referenced.

Reviewer #5, Comment #8: Line 117 - Data on NP independent; Write the full meaning of NP as this is its first usage

• Author Response: Thank you for noting this omission. We have now added this abbreviation.

Reviewer #5, Comment #9: Line 127 - Procedure rates are standardized in 2024 U.S. Dollars. ; Kindly clarify 2024 US Dollars.

• Author Response: We have ensured that we clarify this is 2024 United States (U.S.) Dollars. We have now provided adjusted the introduction to spell out “United States” at its first mention.

o Added Text: “While primary bone cancers account for only 0.2% of primary malignancies in the United States (U.S.)…” (Introduction, Line 67)

Reviewer #5, Comment #10: Line 129 - What does authors mean by bed range, does it mean the number of available beds in a facility? Authors should be very specific in language utilization.

• Author Response: Thank you for this comment. We have now provided a definition of total bed range.

• Added Text:

o “Total bed range was used as a proxy for hospital size and refers to the approximate number of inpatient hospital beds.” (Methods, Lines 141-142)

Reviewer #5, Comment #11: Statistical Analysis. Line 148 - Authors should indicate that mean (SD) or median (IQR) will be used as indicated.

• Author Response: We have now clarified the descriptive statistics section of our methods.

• Added Text:

o “First, a descriptive analysis was conducted to summarize cohort characteristics using frequencies and percentages for categorical variables, and means with standard deviation (SD) or medians with interquartile range (IQR) for continuous variables.” (Methods, Lines 162-163)

Reviewer #5, Comment #12: Inferential tests should also be performed across categories to glean reasonable insights into significant differences across meaningful categories.

• Author Response: While we appreciate the reviewer’s suggestion to perform inferential tests across categories in Table 1, we believe our current multivariate linear regression model already addresses this need in a more statistically rigorous manner. Each regression coefficient (“Estimate”) and its associated p-value represent an inferential test comparing that specific category to the reference group while also controlling for confounders (other hospital, payor, and policy-level variables). This approach avoids the increased risk of Type I errors (false positives) associated with multiple independent hypothesis tests and ensures that the differences reported are not due to confounding factors. Additionally, given that the purpose of the study was not to compare negotiated payor rates or characteristics between the two CPT codes, but rather to provide two examples of CPT codes for bone tumor resections and evaluate them separately, we did not perform inferential testing in Table 1. Therefore, Table 1 is intended purely as a descriptive cohort summary.

Reviewer #5, Comment #13: Authors should specify the assumptions taken to use multivariate linear regression and the parameters used to determine the variables to be included in the model. Did authors check assumptions like multicollinearity, heteroskedascity and others?

• Author Response: We thank the reviewer for this comment. We have updated the Statistical Analysis section to explicitly state the assumptions tested for our multivariate linear regression models. Specifically, we confirmed that our data met the requirements for linearity and normality of residuals. To address the reviewer’s concern regarding multicollinearity, we calculated Variance Inflation Factors (VIF) for all predictors, with all values remaining below a threshold of 5, indicating no significant multicollinearity.

Reviewer #5, Comment #14: What were the model performance metrics used to justify the validity of each of the models such as R squared or adjusted R squared? All these parameters will determine the validity of the model and the validity of the results.

Reviewer #5, Comment #17: Model metrics should be added to the footnotes of all models

• Author Response: We have now added R2 to the footnotes of all the regression tables, which range between 0.19-0.24. This range indicates that the predictors in our model have moderate explanatory power, aligning with our discussion of many factors that contribute to variation in negotiated prices for radical bone tumor resections. These values indicate both that while our selected variables, as expected, do not account for most of the variation, they do explain some of the variation. It also supports the complexity of the pricing market as described in our discussion section.

Reviewer #5, Comment #15: Line 166 - Authors should only report the statistically relevant descriptive statistic should it be median (IQR) or mean (SD). Reporting both is unnecessary.

• Author Response: Thank you for this comment. Given the size of our dataset, we have now removed the median (IQR) and focus solely on reporting the means with standard deviations.

Reviewer #5, Comment #16: the results were mostly repetition of the information on the tables and this should not be so.

• Author Response: Thank you for this important point. We have now significantly revised the results to remove numbers alone within the text and focused on synthesizing the data to allow for an easier read of the results section and a more direct answer to the research question. In response to comment #17, we have also replaced the regression tables with Figures and moved all tables to an eAppendix, which we also believe improves the flow and readability of the results. Consistent with guidelines for writing peer-reviewed manuscripts, we have not provided any interpretation of findings.

• Added Text (examples):

o “Compared to hospitals with 1-100 beds, rates mostly increased with bed count and were the highest in facilities with 1500+ beds (+$797.46, p<0.001).” (Results, Lines 206-208)

o “Payor class was also associated with price variation.” (Results, Line 217)

o “All payor classes had significantly lower rates when compared to the Commercial payor class, with the Medicare advantage class having the lowest rates ($-2946.11, p<0.001).” (Results, Lines 270-272)

Reviewer #5, Comment #17: there was over-utilization of tables; visualisation graphics such as bar plots, box plots and other visualisation means ought to be utilised.

- regression tables can be presented in more explanatory formats with importance placed on models and parameters that explain most of the variance seen in individual models

• Author Response: We thank the reviewer for providing this area for improvement. We have now replaced all the in-text regression tables with Figures (Figures 1-8). The Tables have now been moved as eTables to a supplementary document/Appendix.

Reviewer #5, Comment #18: There should be consistency in font type, size and line spacing with other sections of the manuscript.

• Author Response: Thank you for noting this inconsistency. We have now ensured the entire manuscript is 12-point font and double spaced, including the references.

---

## [Editor Report · Decision Letter 2]

10 Feb 2026

Exploring the Drivers of Price Variation in Orthopaedic Radical Bone Tumor Resection: A Nationwide Database Study

PONE-D-25-44873R2

Dear Dr. Bhat,

We’re pleased to inform you that your manuscript has been judged scientifically suitable for publication and will be formally accepted for publication once it meets all outstanding technical requirements.

Kind regards,

Xiaoen Wei

Academic Editor

PLOS One

Additional Editor Comments (optional):

The authors have satisfactorily addressed the major concerns raised in the previous round of review.
---

## [Editor Report · Acceptance letter]

PONE-D-25-44873R2

PLOS One

Dear Dr. Bhat,

I'm pleased to inform you that your manuscript has been deemed suitable for publication in PLOS One. Congratulations! Your manuscript is now being handed over to our production team.

Kind regards,

on behalf of

Dr. Xiaoen Wei

Academic Editor

PLOS One